# Unique Aspects of Human Placentation

**DOI:** 10.3390/ijms22158099

**Published:** 2021-07-28

**Authors:** Anthony M. Carter

**Affiliations:** Cardiovascular and Renal Research, Institute of Molecular Medicine, University of Southern Denmark, DK-5230 Odense, Denmark; acarter@health.sdu.dk

**Keywords:** decidual reaction, fetal membranes, placental hormones, primates, uterine spiral artery, uterine NK cell

## Abstract

Human placentation differs from that of other mammals. A suite of characteristics is shared with haplorrhine primates, including early development of the embryonic membranes and placental hormones such as chorionic gonadotrophin and placental lactogen. A comparable architecture of the intervillous space is found only in Old World monkeys and apes. The routes of trophoblast invasion and the precise role of extravillous trophoblast in uterine artery transformation is similar in chimpanzee and gorilla. Extended parental care is shared with the great apes, and though human babies are rather helpless at birth, they are well developed (precocial) in other respects. Primates and rodents last shared a common ancestor in the Cretaceous period, and their placentation has evolved independently for some 80 million years. This is reflected in many aspects of their placentation. Some apparent resemblances such as interstitial implantation and placental lactogens are the result of convergent evolution. For rodent models such as the mouse, the differences are compounded by short gestations leading to the delivery of poorly developed (altricial) young.

## 1. Introduction

Adverse pregnancy outcomes can often be linked to defects in placentation [1]. Ethical considerations preclude detailed exploration of the underlying mechanisms. Unfortunately, there are also limitations to what can be learned from animal models. The mouse (*Mus musculus*) and other murine rodents have exceedingly short gestations. Whilst they may be informative about early events, such as the differentiation of cell lineages, they are unsatisfactory for modelling the events of third-trimester human pregnancy [2]. In addition, there are important differences between rodent and human in placentation and the disposition of fetal membranes such as the yolk sac. The objective of this review is to discuss these unique features of human placentation, to define their appearance during the evolution of primates and to contrast them with rodents.

Primary functions of the placenta are gas exchange and the transfer of substrates from mother to fetus. The underlying mechanisms are similar across mammals. Thus the sheep is an excellent model for studying the oxygen supply to the fetus despite structural differences between human and ovine placentation [3]. Similarly, glucose transfer by facilitated diffusion uses the same set of transporters across species [4]. These topics will not be further explored. The interactions between the trophoblast and the maternal immune system are manifold, and a full reckoning cannot be made here. The section on placental immunology therefore focuses mainly on the uterine natural killer (uNK) cells and their ligands. Information on other immune cells, including macrophages, T-cells and innate lymphoid cells, should be sought elsewhere [5,6].

### Mammalian Evolution and Phylogeny

The uniqueness of human placentation can best be assessed in the evolutionary framework provided by phylogenetics. I have striven to keep terminology to a minimum, yet some context is needed, especially for primates. In a broader perspective, eutherian mammals can be sorted into four major clades (Figure 1A). Here we shall deal mainly with one of those clades (Euarchontoglires) and its two subdivisions (Figure 1B). Euarchonta comprises primates, tree shrews and colugos. Glires comprises rodents and lagomorphs. The split between Euarchonta and Glires is estimated to have occurred in the Cretaceous period some 80 million years ago [7]. Therefore, it is not surprising that placentation in the mouse and other rodent models differs in significant respects from human placentation [2].

The primate order has two major subdivisions: Strepsirrhini and Haplorrhini (Figure 2). The former comprises lemurs and lorises with placentation that differs radically from that of humans [10]. In contrast, Haplorrhini, to which our species belongs, was defined by commonalities in fetal membrane development [11]. It includes tarsiers (Tarsiiformes), New World monkeys (Platyrrhini), Old World monkeys and apes (together Catarrhini). Gibbons and great apes (orang-utans, gorillas, bonobo, chimpanzees, and man) comprise the superfamily Hominoidea.

## 2. Early Development

### 2.1. Interstitial Implantation

In most primates, implantation of the blastocyst is superficial. In macaques and baboons, for example, the trophoblast invades the endometrium to establish a placenta, but the developing embryo remains in the uterine cavity. In contrast, the human blastocyst is pulled into the endometrium, which closes above it so that it is completely embedded by the 12th day [13]. The placental bed is underlain by the basal decidua, and the developing embryo is covered by the capsular decidua. Interstitial implantation is a feature shared with the great apes and the gibbons [14,15]. It does occur in rodents, but the process is not identical and has been independently evolved.

Initial penetration of the endometrium is achieved by syncytiotrophoblast [16]. This is formed by fusion of cellular trophoblast to form a multicellular syncytium. The process depends in large part upon syncytins, which are proteins encoded by endogenous retroviral envelope genes that have been incorporated in the genome and exapted to promote cell fusion in the placenta [17]. Humans have two syncytin genes acquired at different timepoints. Whereas *Syncytin-2* occurs in all haplorrhine primates, *Syncytin-1* is found only in apes [17]. Syncytin genes occur in a wide range of mammals, and each represents a separate gene capture [17]. However, it has been argued that the capture of retroviral envelope genes was a prerequisite for the evolution of invasive placentation in mammals [18]. 

### 2.2. Initial Decidual Reaction

The maternal response to implantation is the decidual reaction, which involves the transformation of fibroblast-like endometrial stromal cells into polygonal decidual stromal cells [19]. The decidual reaction once was thought to be absent or atypical in elephants and carnivores [20], yet recent work shows it to be a characteristic feature of eutherian mammals [19]. However, the decidual reaction is transient in some species, such as the nine-banded armadillo (*Dasypus novemcinctus*), and has been lost in many with non-invasive placentation such as cattle (*Bos taurus*) [21]. It is now thought that the decidual reaction evolved from an inflammatory response that is present in marsupials, where it imposes a limit on the length of gestation [22,23]. Several of the genes involved in this response have been downregulated in eutherians, while genes beneficial to implantation have been upregulated [24].

In humans and many primates, as well as rodents, decidual stromal cells persist throughout gestation and have acquired an additional role in pregnancy maintenance [19]. These novel functions arose in the lineage of the large clade Euarchontoglires [19]. 

### 2.3. Early Differentiation of Mesoderm and Secondary Yolk Sac

One of the first fetal membranes to form in mammals is a bilaminar yolk sac comprising an outer layer of trophoblast and an inner lining of the extraembryonic endoderm. It may later acquire blood vessels and function as a choriovitelline placenta. In humans, however, the primary yolk sac is short-lived due to precocious differentiation of the extraembryonic mesoderm, which intrudes between the endoderm and trophoblast (Figure 3). This leads to formation of the secondary yolk sac, which consists of mesoderm and endoderm and becomes a free floating structure within the exocoelomic cavity [25]. Despite lack of contact with maternal tissues, the secondary yolk sac plays an important role in nutrient supply to the first trimester embryo [26]. Precocious development of the extraembryonic mesoderm is a defining feature of haplorrhine primates [11]. Recent work comparing gene expression in the common marmoset (*Callithrix jacchus*), rhesus macaque (*Macaca mulatta*) and human suggests extraembryonic mesoderm is derived in part from the extraembryonic endoderm [27,28]. 

Rodents pursue an entirely different course resulting in an inverted yolk sac with an outward-facing layer of endoderm that persists throughout pregnancy [29]. There are similarities but also marked differences in gene expression and regulatory pathways between the mouse and primates [28].

### 2.4. Allantoic Stalk

The chorioallantoic placenta is formed by the fusion of the allantois with the chorion (trophoblast and extraembryonic mesoderm). In most mammals, the allantois also encloses a fluid-filled space. Indeed, a medium to large allantoic sac is the ancestral state for eutherians [30], and it forms a prominent structure in some species. As an example, cattle have 6–9 litres of allantoic fluid against 2.5 litres of amniotic fluid [31]. In contrast, the human allantois develops as a small diverticulum, and the connection between embryo and placenta, which carries the blood vessels, is the allantoic stalk. This is yet another shared feature that defines haplorrhine primates [11,28]. An allantoic sac is absent in rodents, but this may be due to convergent evolution as it is found in their sister group, the lagomorphs (e.g., rabbit *Oryctolagus cuniculus*) [29].

## 3. Placentation

### 3.1. Haemochorial Placentation

Invasive placentation is the basal condition in eutherians [30,32]. Some strepsirrhine primates (lemurs and lorises) have epitheliochorial placentation, but this is widely regarded as a derived trait [10]. In haemochorial placentas, the trophoblast is in direct contact with maternal blood. Early in human gestation the interhaemal barrier includes two layers of trophoblast [33], but the cytotrophoblast layer (Langhan’s layer) later becomes discontinuous. The interhaemal barrier then comprises syncytiotrophoblast, a thin layer of connective tissue, and the fetal capillary endothelium (Figure 4). Therefore, human placenta is classified as haemomonochorial [34]. Mouse and rat (*Rattus norvegicus*) have three layers of trophoblast, rabbits two, and guinea pigs (*Cavia porcellus*) one, but the significance of these differences should not be overstated [2].

### 3.2. Villous rather Than Labyrinthine Placentation

Of greater consequence is the internal structure of the placenta. Most haemochorial placentas are labyrinthine and organised so that maternal blood channels are arranged in parallel with fetal capillaries. Maternal and fetal blood flow in opposite directions allowing for efficient countercurrent exchange [36,37]. This is also the case in the mouse [38]. In human placenta, on the other hand, the terminal villi are suspended in the intervillous space, which is supplied with blood by the spiral arteries of the basal plate. This is a pattern shared with Old World monkeys and apes [39]. An intermediate form of placentation, where the villi remain connected by bridges of trophoblast (trabeculae), is found in tarsiers and New World monkeys. The evolution of a villous placenta from a labyrinthine one negates the benefit conferred by countercurrent exchange. However, opening up the maternal component allows for a larger volume flow of blood and, thus, greater oxygen delivery, and this likely outweighs the loss of countercurrent exchange [4].

### 3.3. Uterine Spiral Artery Transformation

A key feature of human placentation is the transformation of the uterine spiral arteries to wide vessels with low resistance to flow. An initial phase involves vacuolation and focal loss of endothelial cells and loosening of the smooth muscle layer. It appears to be dependent on cytokines secreted by uNK cells and macrophages [40,41,42]. The second phase is associated with invasion of the endometrium and vessel walls by extravillous trophoblast. This leads to complete loss of endothelium and disruption of the smooth muscle with the greatly widened vessels eventually being lined by trophoblast embedded in a fibrinoid layer [41]. 

With advancing pregnancy there is also dilatation of the radial arteries, arcuate arteries and uterine arteries, none of which are invaded by trophoblast. This is most likely due to stimulation by oestrogens and nitric oxide-mediated flow-dilation signals [43].

### 3.4. Trophoblast Invasion by Interstitial and Intravascular Routes

In human pregnancy, the trophoblast invades by two routes [16]. Firstly, it migrates from the basal plate into the lumina of the uterine spiral arteries against the direction of flow (the intravascular route). Secondly, the trophoblast differentiating from the anchoring villi migrates through the decidua towards the blood vessels (the interstitial route). In a healthy pregnancy, trophoblast invasion extends through the relatively shallow endometrium to the inner third of the myometrium. Shallower invasion leads to inadequate transformation of the spiral arteries, thereby limiting the blood supply to the intervillous space, and is causally associated with fetal growth restriction and preeclampsia [1,44]. Trophoblasts that invade by the interstitial route undergo endoreduplication [45] and go no deeper than the inner myometrium, where they are found as multinucleate giant cells [46]. 

Our studies in chimpanzee and gorilla suggest that the depth of trophoblast invasion and spiral artery transformation resembles the human condition [47,48]. In Old World monkeys, however, there is rapid invasion of spiral arteries by the intravascular route but none by the interstitial route; indeed, there is a sharp border between the cytotrophoblastic shell and decidua [49,50]. This also appears to be the case in gibbons, suggesting that invasion by the interstitial route evolved in the lineage of the great apes [51]. In Old World monkeys, intravascular trophoblast is confined largely to the endometrial segments of the spiral arteries [49]. The situation in New World monkeys and tarsiers is not sufficiently known [16]. 

Rodent models do not readily conform to any of these features. The depth of trophoblast invasion in rodents varies but can extend to the mesometrial arteries, as in the guinea pig [52]. In the mouse, trophoblast glycogen cells migrate to the decidua but do not invade its vessels [53]. Instead, trophoblasts of the giant cell lineage migrate to and line the spiral arteries [54]. However, this does not occur until vascular remodelling is complete [5,55]. Indeed, rodents are unsatisfactory models of trophoblast invasion. Thus, deeper penetration of the arteries was found in a rat model of preeclampsia [56], which is the opposite of the shallower invasion typical of preeclampsia in human pregnancy [1].

## 4. Immunology of Decidua and Trophoblast

The placenta is a semi-allograft, yet it is not rejected by the maternal immune system. This immunological paradox was framed by Sir Peter Medawar [57] and remains a pivotal question in reproductive immunology [6]. As many as 70% of leukocytes in the uterus are uNK cells [58]. Their properties differ from those of peripheral natural killer cells and include lower cytotoxic activity [42]. Their putative role in the early stages of uterine artery transformation, alluded to above, may be explained by the secretion of cytokines, growth factors and proteases [40,42]. Here, we are concerned with their interplay with invasive extravillous trophoblast. 

### Interplay of uNK Cell Receptors and HLA Antigens

Human trophoblast does not express the major histocompatibility antigens (MHC) Class I, which include human leukocyte antigens (HLA) A and B. Instead, the surface of trophoblast presents HLA-C, which exhibits a high degree of allelic polymorphism, as well as HLA-E and HLA-G. There is no equivalent to HLA-C in monkeys or gibbons [59,60], but an invariant form appears in orangutans [61]. A later gene duplication yielded the two epitopes C1 and C2, which are found in chimpanzees, gorillas and humans [62].

HLA-C1 and -C2 are the principal ligands for the killer immunoglobulin-like receptors (KIRs) on uNK cells. KIRs likewise exhibit a high degree of allelic polymorphism, and importantly, there are inhibitory and excitatory variants [63]. Thus, numerous combinations are possible of HLA-C presented by the trophoblast (with one allele being paternal in origin) and KIRs expressed by maternal uNK cells. This can affect pregnancy outcome. When the trophoblast expresses HLA-C2 and the uNK cells express only inhibitory receptors, the combination is associated with a higher incidence of recurrent abortion, fetal growth restriction and preeclampsia [64,65]. KIR genes have evolved along separate pathways in great apes and human [66]. Indeed, it has been proposed that the emergence of the HLA-C2 epitope in apes is causally linked to the advent of preeclampsia. This is difficult to prove as reports of eclampsia in great apes are largely anecdotal (see [51]). In any case, the evolution of KIRs has pursued different paths in nonhuman primates [67]. 

HLA-G is expressed exclusively on trophoblast and has been implicated in maternal immune tolerance [68,69]. *MHC-G* is expressed in the great apes [70] but is a pseudogene in baboons (*Papio* spp.), rhesus macaque (*Macaca mulatta*), cynomolgus macaque (*M. fascicularis*) and vervet monkey (*Chlorocebus aethiops*), where its function is assumed by a new gene *MHC-AG* [60,71]. Receptors for HLA-G and MHA-AG are expressed by uNK cells and include KIR2DL4 [60].

The uterus of rodents is also rich in uNK cells, and they appear to be essential for the transformation of vessels analogous to human spiral arteries [72,73]. However, the principal receptors on rodent uNK cells belong to the lectin-like family (Ly49) [74], so rodents are not useful for exploring interactions between KIRs and HLA antigens. Rodents do not have MHC-G, although *HLA-G* expression has been achieved in transgenic mice [75].

## 5. Endocrinology of the Placenta

Placental hormones are secreted to the maternal circulation and adapt maternal physiology to meet the requirements of pregnancy and subsequent lactation [76]. Many of the peptides made by human trophoblast are unique to the primate lineage. Placental lactogens occur in human and rodents but have arisen through convergent evolution and serve different functions. Pregnancy maintenance depends on progesterone secretion, but an unusual feature of human pregnancy is that parturition occurs without a fall in plasma progesterone.

### 5.1. Chorionic Gonadotrophins

Human chorionic gonadotrophin (hCG) is responsible for early pregnancy maintenance. It evolved through duplication of the gene encoding the β-subunit of luteinising hormone. This occurred in the lineage of haplorrhine primates followed by further duplications so that many primates have multiple genes and pseudogenes [77]. A chorionic gonadotrophin was convergently evolved in the lineage of equids [78].

### 5.2. Placental Lactogens and Growth Hormones

Human placenta also secretes a placental lactogen, hPL, previously known as chorionic somatomammotropin. The genes that code for hPL are derived from the growth hormone gene [79]. A third gene in the cluster codes for placental growth hormone, which supplants pituitary growth hormone in the latter part of pregnancy [80]. All haplorrhine primates have placentally expressed genes related to growth hormone but there is great variation especially between New World and Old World monkeys [81,82], which may have attained placental expression separately [83]. 

The placental lactogens of muroid rodents, PL1 and PL2, are responsible for the maintenance of the corpus luteum [84]. In contrast to primates, they were derived by duplication from the prolactin gene together with a range of other cytokines [84]. Thus, placental lactogens of primates and rodents differ both in origin and function.

### 5.3. Progesterone and Its Receptors

Pregnancy maintenance in mammals requires the presence of progesterone secreted from the corpus luteum or placenta [85]. In many species, parturition is initiated through a fall in circulating progesterone, so-called progesterone withdrawal. In humans, where the placenta synthesises progesterone from maternal cholesterol, secretion is maintained right up to the start of labour [86]. This contra-intuitive finding led to the concept of a “functional” progesterone withdrawal for which the favoured explanation focusses on progesterone receptors (PR) in the myometrium. Of the two major isoforms, PR-B is the stronger trans-activator of progesterone-responsive genes, and PR-A acts as a trans-suppressor of PR-B’s effect. They are equally expressed in the myometrium throughout gestation. However, parturition is associated with a change in the PR-A to PR-B ratio due to increased expression of PR-A [87]. This switch contributes to myometrial activation via the activator protein-1 (AP-1) pathway [88]. Interestingly, there is evidence for adaptive evolution of the progesterone receptor gene (*PGR*) in the human lineage [89,90]. Of note, PRs are also expressed in human decidua, and a recent hypothesis points to a decline in decidual PR expression as a possible factor in the initiation of parturition [91].

Plasma progesterone levels increase before parturition in the rhesus macaque [92], and there is evidence of rapid evolution of *PGR* in catarrhine primates [90].

In mouse and rat, the corpus luteum is the sole source of progesterone, and plasma concentrations fall precipitously before parturition. These models are of limited value in understanding human parturition [93]. The situation is different in hystricomorph rodents: the placenta is a major source of progesterone in the guinea pig, and there is no change in circulating progesterone prior to parturition [86]. Therefore, it is considered a more appropriate model for parturition research [93].

## 6. Pregnancy Duration and Newborn State

An undeniably unique feature of human reproduction is that newborn babies are helpless and entirely dependent on parental care [94]. They differ to some degree from other haplorrhine primates, although a long childhood is a general feature of great apes (Table 1).

Mammals tend to adopt one of two contrasting strategies [108]. In the first, a short gestation with a large litter leads to the birth of poorly developed or altricial offspring. In the second, a long gestation with a small litter (usually singleton) ends with the birth of well-developed offspring with open eyes and ears, a coat of hair, and some degree of independence. Human babies have most of the attributes of precocial offspring but are helpless at birth and require parental care for several years. 

The human pelvis has been remodelled to enable bipedal walking. Therefore, there has been a trade-off between prenatal brain development, i.e., the size of the fetal head, and the diameter of the birth canal [109]. As a result, the volume of the brain at birth is about a quarter of adult size compared to 40% in the chimpanzee [110]. Indeed, in humans, the fetal pattern of brain growth continues for a year after birth [94]. Consequently, the fontanelles separating the bones of the skull do not close until 18 months to 2 years after birth [111]. The development of most other organs is as complete at birth as in other primates, all of which deliver precocial young.

In contrast, rodents such as mouse and rat deliver truly altricial young with closed eyes, naked skin, and incomplete development of major organs, such as the kidneys [2]. The short gestation means that there is no period equivalent to the third trimester of human pregnancy when obstetric complications are most evident. Differences in gestation length are even reflected in placental function, the different role of placental lactogens in rodents and primates being a case in point.

## 7. Discussion

### 7.1. Placental Evolution

Placental characters shared with all eutherian mammals are invasive placentation and the decidual reaction (Table 2). Persistence of decidua into late gestation is common to the major clade Euarchontoglires, which includes rodents as well as primates. Many characteristics are shared with the primate suborder Haplorrhini, including features of the fetal membranes that Hubrecht used to justify classing tarsiers with monkeys and apes [11]. Characters shared with Old World monkeys include villous placentation with an intervillous space and some aspects of trophoblast invasion. However, implantation is superficial in all primates except gibbons and great apes, and trophoblast invasion by the interstitial route is shared only with the great apes. This leads to the evolutionary timeline shown in Table 2.

### 7.2. Pregnancy Complications

Comparatively little is known about pregnancy complications in nonhuman primates. Preeclampsia may occur in great apes, but the evidence is thin, although in one case supported by a renal biopsy [113]. There is, however, much to be said for the argument that deep trophoblast invasion, especially by the interstitial route, can be linked to the emergence of preeclampsia in the great apes [51]. 

Many changes reminiscent of preeclampsia could be replicated in a baboon model by uterine artery ligation [114], but these may merely reflect responses to reduced oxygen delivery. Hypertension can develop spontaneously in the vervet monkey, even when not pregnant [115], and gestational hypertension in the closely related patas monkey (*Erythrocebus patas*) was accompanied by preeclampsia-like symptoms [116]. 

Fetal growth restriction is another focus of obstetric research. It occurs in New World monkeys that regularly bear twins or triplets, and the effects on birth weight and neonatal outcomes are currently under investigation [117]. Indeed, the common marmoset (*Callithrix jacchus*) is a promising model for pregnancy research [2].

## 8. Conclusions

The many unique features of human pregnancy and placentation pose problems in planning and interpreting animal experiments. Two factors are involved. The first is phylogenetic distance. Quite a few features are shared with haplorrhine primates, making them the models of choice. Baboons and macaques share additional features such as endovascular trophoblast and spiral artery transformation as well as a true intervillous space. On the other hand, maintenance of breeding colonies is costly. Therefore, it is worth considering the common marmoset for which caging and feeding costs are much lower [118]. As mentioned in the introduction, primates and rodents last shared a common ancestor in the Cretaceous period, so it is not surprising that placental evolution has pursued different paths. There has, for example, been convergent evolution of placental lactogens to serve different purposes.

A second factor compounds the problem with rodent models. This is the difference in reproductive strategies. The short generation times of mice and rats make them ideal laboratory animals. Unfortunately, the same qualities render them unsatisfactory for pregnancy research [119,120]. The major obstetric syndromes become manifest in the third trimester, but there is no equivalent period in the mouse. The newborn mouse is truly altricial, with much of organ development occurring in the postnatal period. I have been at pains to stress, as argued by Martin [111], that human babies are precocial in almost all aspects save brain development; they are not altricial. Alternative rodent models are the spiny mouse (*Acomys cahirinus*) and guinea pig, both of which deliver precocial young [2].

## Figures and Tables

**Figure 1 ijms-22-08099-f001:**
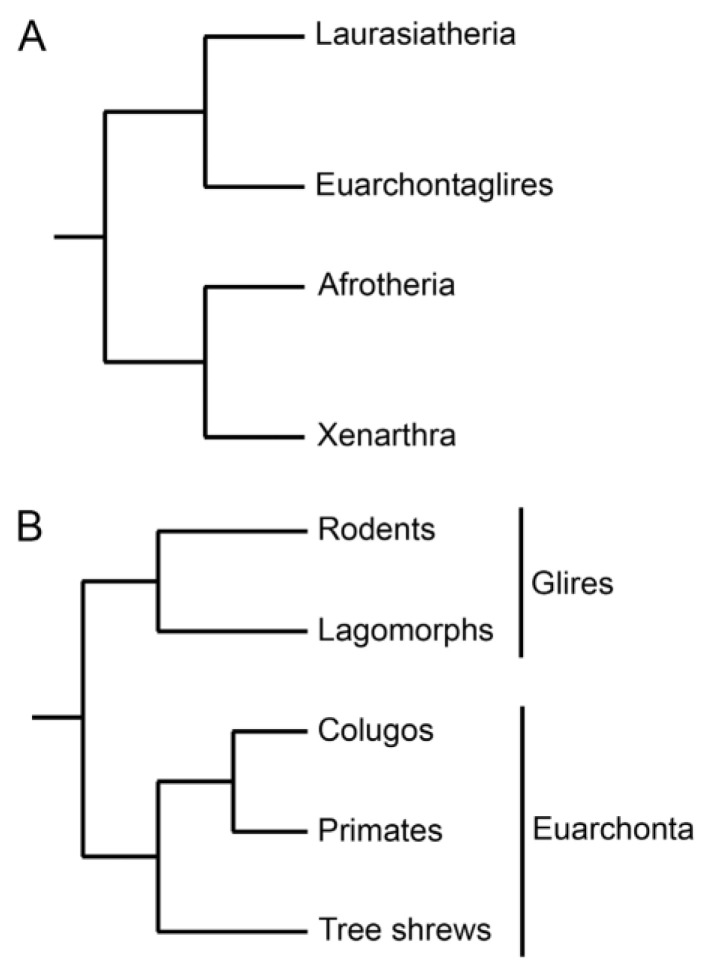
The mammalian tree. (**A**) The four major clades of eutherians [8]. (**B**) The orders of Euarchontoglires [9]. Note the separation of Glires (including rodents) from Euarchonta (including primates). There are alternative interpretations of the root of the tree and the position of tree shrews. Reprinted with permission from [2] © 2021 Society for Reproduction and Fertility.

**Figure 2 ijms-22-08099-f002:**
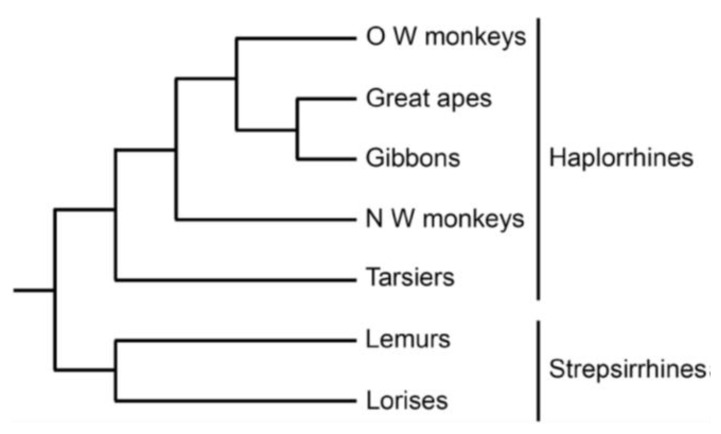
Classification of primates [12]. Strepsirrhines and haplorrhines are suborders, tarsiers are regarded as an infraorder, whilst the other clades shown are superfamilies. OW, Old World; NW, New World. Reprinted with permission from [2] © 2021 Society for Reproduction and Fertility.

**Figure 3 ijms-22-08099-f003:**
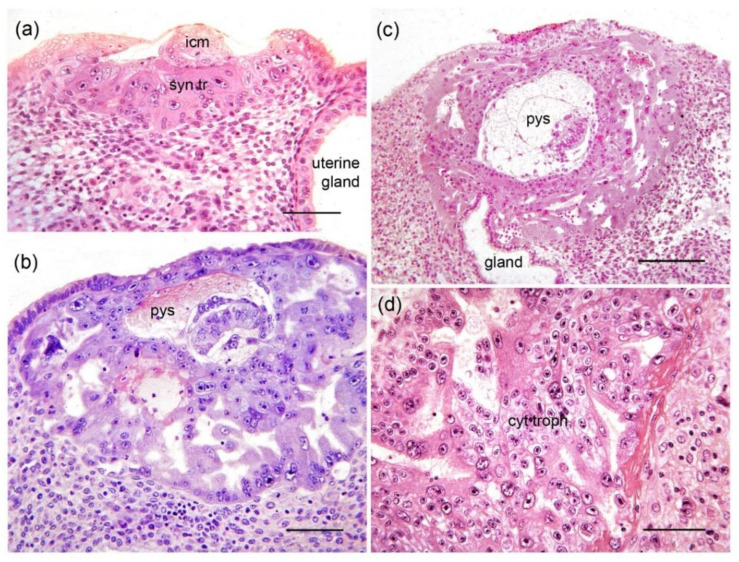
Early differentiation of mesoderm in the human embryo. (**a**) Trophoblastic plate stage (Carnegie Stage 5a) showing the inner cell mass (icm). The cavity of the blastocyst is collapsed. The pad of trophoblast has different sized nuclei and both cellular and syncytial trophoblast (syn tr) (Carnegie Embryo #8020). Scale bar, 70 µm. (**b**) Early lacunar stage (Carnegie Stage 5b) Some maternal blood has leaked into the primary yolk sac (pys). Note the irregular shape of the lacunae on the right, which appear to be formed from expanding clefts (Carnegie Embryo #8004). Scale bar, 90 µm. (**c**) Lacunar stage (Carnegie Stage 5c). Note the anastomotic lacunae within the syncytiotrophoblast. In the area between the trophoblast and the already partially constricted primary yolk sac (pys), there are mesenchymal cells (extraembryonic mesoderm) (Carnegie Embryo #7699). Scale bar, 176 µm. (**d**) Predecessors of the primary villi (Carnegie Stage 6). The cytotrophoblast (cyt troph) is accumulating in the partitions between lacunae, initiating the formation of primary villi (Carnegie Embryo #9260). Scale bar = 70 µm. Reprinted with permission from Carter, Enders and Pijnenborg [16] © The Authors. Published by the Royal Society. All rights reserved.

**Figure 4 ijms-22-08099-f004:**
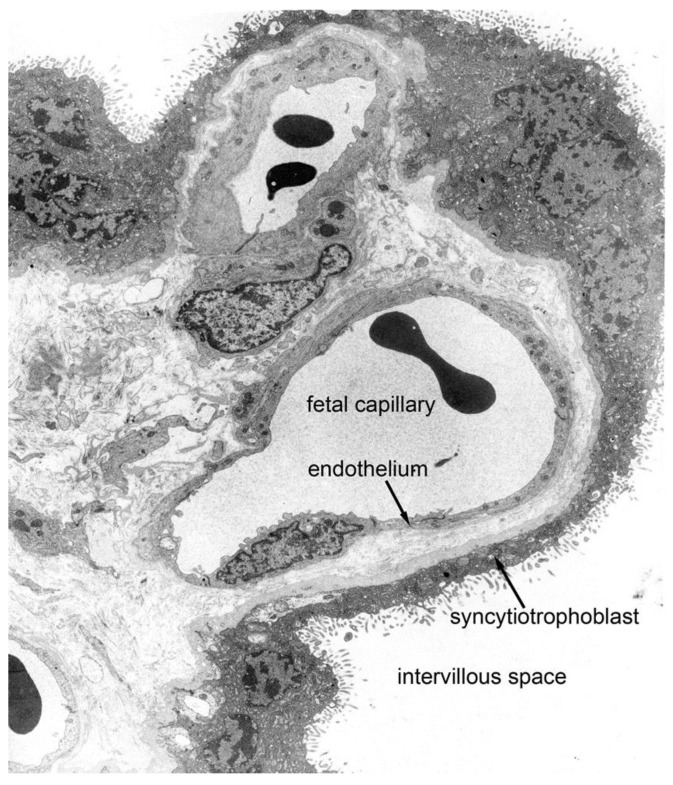
The interhaemal barrier of the human placenta is classified as haemomonochorial. The intervillous space is separated from blood in the fetal capillary by syncytiotrophoblast and fetal capillary endothelium with their basal membranes. A very thin layer of connective tissue cytoplasm is interposed between the two basal membranes. Courtesy of Dr. Allen C. Enders. Reproduced with permission from [35] Copyright © American Physiological Society.

**Table 1 ijms-22-08099-t001:** Precocity and parental care in selected primates. Apart from tarsier and marmoset, data are from field observations on free-living populations. Gestation lengths are approximate and based on few observations.

Clade	Species	Common Name	Length of Gestation	Parental Care	References
Tarsiers	*Cephalopacus bancanus*	Western tarsier	178 days	Nutritional and social independence by 60 days	[95]
New World monkeys	*Callithrix jacchus*	Common marmoset	143–144 days	Independent movement by 3 weeks; weaning by 3 months	[96,97]
Old World monkeys	*Papio cynocephalus*	Yellow baboon	178 ± 6 days	Milk supplemented early with plant foods; fully weaned after about a year; carried for 8 months	[98,99,100]
Lesser apes	*Symphalangus syndactylus*	Siamang	230–235 days	Partial weaning at 6 months; travel independently by 1 year	[101]
Great apes	*Pongo pygmaeus*	Bornean orangutan	275 days	Partial weaning by 11 months; fully independent at 7–10 years	[102,103]
	*Gorilla beringei*	Eastern gorilla	255 days	Weaning at 3–4 years	[103,104,105]
	*Pan troglodytes*	Chimpanzee	196–260 days	Weaning at 10 months; dependent on mother for 5 years	[103,106,107]

**Table 2 ijms-22-08099-t002:** Characteristics of human placentation and their estimated appearance during evolution. Branching points (Mya, million years ago) are estimates based on molecular data [7,112].

Character	Taxonomic Clade	Branching Point	Geological Period or Epoch	Comments
Invasive placentation	Eutheria	98.5 Mya	Late Cretaceous	
Decidual reaction	Eutheria	98.5 Mya	Late Cretaceous	An inflammatory response in marsupials
Persistence of decidual stromal cells	Euarchontoglires (includes rodents and primates)	91.8 Mya	Late Cretaceous	
Precocious extraembryonic mesoderm	Haplorrhini	44.8 Mya	Middle Eocene	
Secondary yolk sac	Haplorrhini	44.8 Mya	Middle Eocene	
Allantoic stalk	Haplorrhini	44.8 Mya	Middle Eocene	Many mammals have an allantoic sac
Haemomonochorial placentation	Haplorrhini	44.8 Mya	Middle Eocene	
Syncytin-2 *env* gene	Haplorrhini	44.8 Mya	Middle Eocene	
Chorionic gonadotropin	Haplorrhini	44.8 Mya	Middle Eocene	
Placental lactogens and growth hormone	Haplorrhini	44.8 Mya	Middle Eocene	Vary between primate lineages
Trophoblast invasion by intravascular route	Old World monkeys and apes	29.8 Mya	Oligocene	
Villous placentation with an intervillous space	Old World monkeys and apes	29.8 Mya	Oligocene	Trabecular placentation in tarsiers and NW monkeys
Interstitial implantation	Lesser and greater apes	20.2 Mya	Early Miocene	
Syncytin-1 *env* gene	Lesser and greater apes	20.2 Mya	Early Miocene	
Trophoblast invasion by interstitial route	Great apes	15.1 Mya	Middle Miocene	
HLA-C	Great apes	15.1 Mya	Middle Miocene	

## Data Availability

No new data were created or analyzed in this study. Data sharing is not applicable to this article.

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
