# Peer review of "Unique Aspects of Human Placentation"

_ijms, 2021, doi:10.3390/ijms22158099_

Round 1

Reviewer 1 Report

Dear Author, 

I like you paper and the significancy of its topic. It is actual. Generally, I have not any comments for correction, but have few suggestions:

Could you provide more explanation about the uterine artery and its modification in pregnancy? It is so important for physiology of pregnancy. Please, add more detailed background of this process.

Also, if it is possible, please add one image of real placenta at least. 

Thank you. 

Author Response

Response to Reviewer #1

I like you paper and the significancy of its topic. It is actual. Generally, I have not any comments for correction, but have few suggestions:

Thank you for your encouragement.

Could you provide more explanation about the uterine artery and its modification in pregnancy? It is so important for physiology of pregnancy. Please, add more detailed background of this process.

I was a little unsure whether you meant the main uterine artery or the uterine spiral artery. I added a short paragraph about the former. The many cytokines putatively involved in spiral artery transformation are too numerous to mention. Further, in the present context we lack knowledge about non-human primates that would enable identification of unique features.

Also, if it is possible, please add one image of real placenta at least. 

I have reproduced a set of images previously published as Supplementary Figure and therefore little seen (Figure 3). It illustrates the early differentiation of the extraembryonic mesoderm. In addition, there is TEM of the interhaemal barrier at term (Figure 4).

Reviewer 2 Report

The review article (ijms-1300612) entitled “Unique Aspects of Human Placentation” by Carter AM. submitted to International Journal of Molecular Sciences aims to evaluate the current information and relevant mechanisms related to the placental evaluation.

Please see my comments in below:

  • The author should give a short informative information related to the main function/role of placenta in first paragraph in the introduction section, later he should make a comparation between human and rodent placentation.
  • To clarify the readers, the author could add a figure of human and mice placentas displaying similarity and differences between two species.
  • Page 1, Line 22, “the major obstetrical syndromes” should be replaced with adverse pregnancy outcomes.
  • Page 7, Section 5.3; the author should add a sentence why progesterone and progesterone receptor are important for pregnancy, and add which cells are express progesterone receptor in the placenta. In the manuscript, the author discussed only myometrial PR, should also add a information of decidua, because the decidual cells are only PR expressing cells at the maternal-fetal interface. The author should discuss other factor/mechanisms(s) that cause functional P4 withdrawal in myometrium and decidua, such as PR levels, other transcriptional coactivator/repressor and inhibition of PR transcriptional activity.
  • The author should add a small section explaining the selective mechanisms that may contribute to placental diversity.
  • Page 1, Lines 24-25, the author should explain why he used that rodent deliver immature pups. Please clarify the sentence.
  • The author should add a column displaying gestational length in Table 1.

Author Response

The author should give a short informative information related to the main function/role of placenta in first paragraph in the introduction section, later he should make a comparation between human and rodent placentation.

Comparison with rodent placentation is a recurring theme and I feel it important to stress this from the get-go.

To clarify the readers, the author could add a figure of human and mice placentas displaying similarity and differences between two species.

Difficult to conceive of a figure that would cover all differences. Suggestion appreciated but not followed.

Page 1, Line 22, “the major obstetrical syndromes” should be replaced with adverse pregnancy outcomes.

Suggestion followed.

Page 7, Section 5.3; the author should add a sentence why progesterone and progesterone receptor are important for pregnancy, and add which cells are express progesterone receptor in the placenta. In the manuscript, the author discussed only myometrial PR, should also add a information of decidua, because the decidual cells are only PR expressing cells at the maternal-fetal interface. The author should discuss other factor/mechanisms(s) that cause functional P4 withdrawal in myometrium and decidua, such as PR levels, other transcriptional coactivator/repressor and inhibition of PR transcriptional activity.

Thank you for these suggestions. I have followed them in part but do not wish to get into too much detail since the mechanisms have not been sufficiently explored in non-human primates. Therefore, it is difficult to place them in the context of the uniqueness of human placentation. However, I do now refer to the possible role of decidual PR receptors.

The author should add a small section explaining the selective mechanisms that may contribute to placental diversity.

One can only speculate as to the selection pressures. I once tried to answer a similar question specifically about evolution of epitheliochorial placentation (Ann Rev Anim Biosci 2013; 1: 443-467). Having read through it, I am not too keen on repeating the exercise.